# Optimal Design and Control Performance Evaluation of a Magnetorheological Fluid Brake Featuring a T-Shape Grooved Disc

Pacifique Turabimana [1] and Jung Woo Sohn [2,*]

1 Department of Aeronautics, Mechanical and Electronic Convergence Engineering, Graduate School, Kumoh National Institute of Technology, Daehak-Ro 61, Gumi 39177, Republic of Korea; 20215056@kumoh.ac.kr
2 Department of Mechanical Design Engineering, Kumoh National Institute of Technology, Daehak-Ro 61, Gumi 39177, Republic of Korea
* Correspondence: jwsohn@kumoh.ac.kr

**Abstract:** Magnetorheological fluid brakes are a promising technology for developing high-performance drive-by-wire braking systems due to their controllability and adaptability. This research aims to design an optimal magnetorheological fluid brake for motorcycles and their performance. The proposed model utilizes mathematical modeling and finite element analysis using commercial software. Furthermore, the optimization of this MR brake is determined through multi-objective optimization with a genetic algorithm that maximizes braking torque while simultaneously minimizing weight and the cruising temperature. The novelty lies in the geometric shape of the disc, bobbin, and MR fluid channels, which results in a light MR brake weighing 6.1 kg, an operating temperature of 89.5 °C, and a power consumption of 51 W with an output braking torque of 303.9 Nm. Additionally, the control performance is evaluated using an extended Kalman filter controller. This controller effectively regulates braking torque, speed, and slip rate of both the rear and front wheels based on road characteristics and motorcycle dynamics. This study's findings show that the front wheel necessitates higher braking torque compared to the rear wheel. Moreover, the slip rate is higher on the rear wheel than on the front wheel, but the front wheel stops earlier than the rear wheel.

**Keywords:** magnetorheological fluid; T-shape disc; motorcycle brake; extended Kalman filter

## 1. Introduction

The automotive industry is developing continuously to initiate safer, cheaper, and faster systems and components for better vehicle performance. Advanced technology also takes place in this industry sector. Automotive mechanical systems and actuators like suspension, clutch, brake, gearbox, cooling systems, and others are replaced by electromechanical systems, which use advanced technology. Electromechanical systems present many advantages, not limited to being environmentally friendly, having high operation safety, being easily controllable, and having higher performance. The trending technologies for these actuators are drive-by-wire technologies that use magnetorheological (MR) and electrorheological (ER) principles [1,2]. ER and MR fluids are smart materials that belong to controllable fluids, and both were discovered in the 20th century [3]. However, MR fluids are the most used and have numerous advantages over ER fluids. Some of these advantages include small electric power consumption, short response time, and not being sized largely [4,5]. Nguyen et al. [6] stated in their research that MR fluids contain micron-sized ferromagnetic particles which can be magnetized, changing the rheological properties of the fluid with the application of magnetic fields when there is the presence of electricity. Thus, MR-fluids-based automotive actuators operate with four main operation modes: flow mode, squeeze mode, shear mode, and valve mode [7,8]. Different automotive actuators

were studied by various researchers. Furthermore, the successful use of this technology of magnetorheological fluid is not limited to MR engine mountings [9], MR clutches [10], MR dampers [11], and MR brakes [6,12].

Recently, several researchers of MR fluid application in the automotive field were interested in replacing conventional actuators with MR fluid actuators. They have conducted much research in terms of simulations and experiments. However, an MR brake replaces the conventional hydraulic (CH) brake system. Hence, the MR brake is critical to improving vehicle performance by increasing its safety assurance rate. CH brake systems present different operating problems, such as having a long response time of between 200 and 300 milliseconds, being unfriendly to the environment, braking noise, high temperature from the contact of metal and lining materials, non-controlled and irregular output braking torque, requiring many system components, difficulty in distributing and balancing torque to all vehicle wheels, and having insufficient braking outputs at high wheel speeds [1,12]. Contrarily, MR brakes present many advantages which can overcome the operating difficulties of conventional hydraulic brakes. Within operation, MR brakes are faster in their time response; have a high, controllable torque output; require little space for the system components; have a simple mechanism; and have braking performance without metal-to-metal friction [1,8,13–15].

Various research works on MR brakes present the theoretical torque outputs from different models but do not focus on the time delay to generate torque, amount of electric power consumption, model structure, operational optimization, and performance evaluation of the MR brake. A couple of researchers of MR brakes came up with technical approaches by focusing on the highlighted properties. Thang et al. [16] designed an MR brake with a single disc using a differential evolution algorithm for geometric optimization. In the optimization process, the design variables were considered as discretized variables. Hadi and Ramin [17] proposed an MR brake applied in automotives, and they evaluated its performance. The model was optimized and controlled to fit the quarter vehicle under different road conditions. The optimal braking outputs were achieved after optimizing the model using the method of multi-objective optimization with a combination of genetic and sequential quadratic programming algorithms. The control of the vehicle's wheel lock-up used a PID controller by regulating the applied electric current. Nguyen et al. [18] proposed an MR brake with a disc-type model structure to replace the conventional hydraulic brake of the motorcycle's front wheel. The output braking torque was calculated and analyzed based on the Bulkley Herschel rheological model of MR fluids at a maximum of 260 Nm at 5A of supplied electric current. The model structure was optimized to maximize the braking torque and minimize the operating temperature generated in the MR fluid. Nguyen and Choi [19] evaluated the performance of a T-shaped drum-type MR brake by applying it to a mid-sized motorcycle. Their research work is limited to the measured torque output of the MR brake to replace the hydraulic brake of the rear wheel of the mid-sized motorcycle. The maximum torque generated by this brake is 450 Nm with 2A of electric current. It does not include the motorcycle dynamics and driving road conditions required for braking systems. In the research work done by Sohn et al. [1], a disc-type MR brake was optimally designed and experimented with. Its performance was evaluated by using the MR brake to replace the CHB for the front wheel of a mid-sized motorcycle. The designed model was geometrically optimized using a gradient descent algorithm that is integrated with ANSYS software. The conducted experiments resulted in 260 Nm maximum output torque with 3A of the electric current supplied to the excitation coil. The results of the experiments are the same as the simulation results from the optimized model recorded from ANSYS software. Thang et al. [20] conducted research work on designing and experimenting with the MR brake with a tooth-shaped disc for small-size motorcycles. In the optimization process of the proposed model, they parameterized the model to maximize braking torque by minimizing the mass of the MR brake. With the PID controller, torque response was obtained, but it was tuned manually. There is a difference of 11% between the output torque from experiments and the optimized simulation results.

The maximum simulation braking torque is 150 Nm while the experiment's output torque is 133.5 Nm with a supply coil of 2.5 A of electric current. Wang and Choi [21] proposed an MR brake with a fuzzy logic sliding mode (FLSM) controller to achieve higher safety and steering stability. The proposed MR brake has been applied in the automotive industry during its performance evaluation. The FLSM controller includes vehicle wheel dynamics and road characteristics during the control of braking outputs torque, and the wheel can be locked up to operate as an anti-lock braking system (ABS).

In this study, we aim to design the optimal T-shape grooved disc magnetorheological fluid brake (T-SGDMRB) for motorcycle braking. The braking torque calculations utilize the Bingham plastic model for MR fluids, considering the radial and annular MR fluid gaps. Finite element simulations using COMSOL Multiphysics evaluate the magnetic field intensity generated in MR fluid gaps concerning the supplied current. The geometric parameters of the MR brake undergo design optimization using a multi-objective optimization with a genetic algorithm (GA) method. The optimization seeks to increase braking torque while minimizing the weight of the MR brake and heat generated during braking, thus improving the safety and performance of the motorcycle (BMW R1200RT). Subsequently, brake actuation performance control for the proposed optimally designed MR brake is conducted using a simulation approach with MATLAB software. In the performance evaluation of the designed MR brake, we consider both road conditions and dynamic behaviors of the motorcycle, and we control wheel slip using an extended Kalman filter. The stopping time and distance of each wheel of the motorcycle have been calculated numerically by considering different riding speeds, with two MR brakes on the front wheel and a single MR brake on the rear wheel.

## 2. Development of a Proposed MR Brake

### 2.1. Structure Configuration of Proposed MR Brake

The geometric structure defines the shape and arrangement of components for the proposed MR brake. Figure 1a illustrates the 2-D axisymmetric structure, indicating the position of each part of the T-SGDMRB. The disc serves as the base structure for other components and MR fluid paths within the brake. Figure 1b shows the five main components of the T-SGDMRB: the disc hub, disc, bobbin, coil, and stator (housing). These components are categorized as either magnetic or non-magnetic.

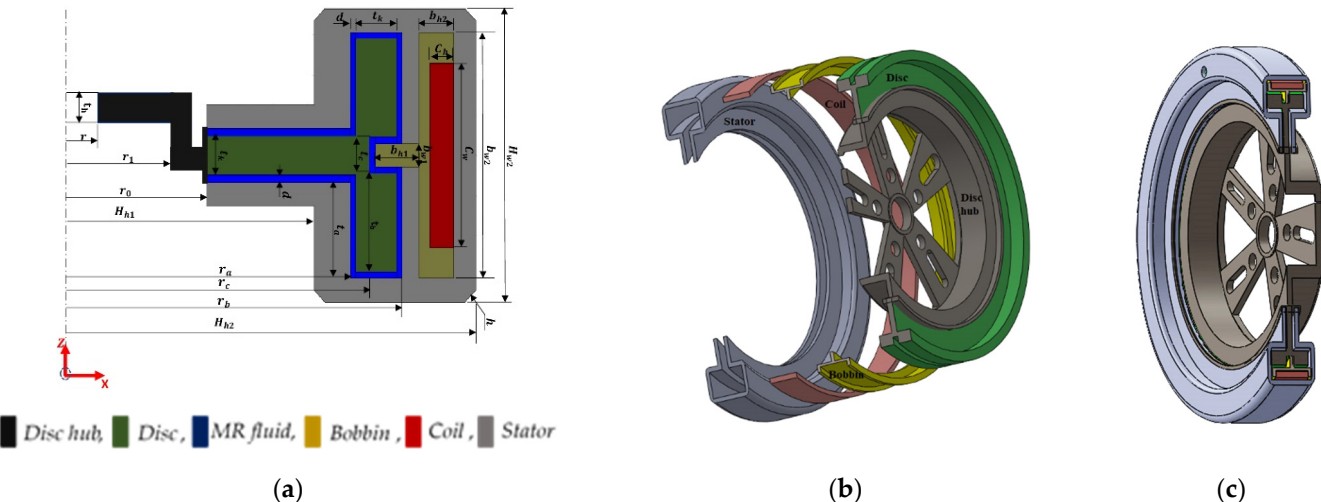

**Figure 1.** Geometric structure of T-shape grooved disc MR brake. (**a**) Axisymmetric with material distribution, (**b**) Main components of proposed MR brake, (**c**) Complete assembled MR brake.

The disc hub is a non-magnetic stainless steel 316 rotating part designed to support and connect the MR brake with the motorcycle's wheel hub, transmitting braking torque from the disc to the wheel hub. The disc, made from magnetic low carbon steel 1010, is

fixed onto the hub and rotates at the same speed as the motorcycle wheel hub. It consists of three parts: a disc leg, a disc flange, and a groove space. The bobbin, made from non-magnetic aluminum, has a two-teethed fork shape. Its bottom part occupies the grooved space of the disc, while the upper part holds the coil, which is a sequence of copper wires wound around the bobbin using standard wire gauge 21 (A.W.G 21 with 0.7230 mm). The stator (housing) is a fixed part of the brake designed to house all static components, such as the coil, bobbin, and a refilling hole for the MR fluid. It is made from low-carbon steel 1008. A sealing ring is placed between the housing and the disc hub to prevent leakage. The MR fluid (MRF-132 DG) is filled in the volume between the housing and disc, as well as between the disc's grooved part and the bobbin. The gaps of MR fluid surrounding the disc leg form a radial channel, while the remaining gaps create an annular channel.

### 2.2. Mathematical Modeling of Torque for T-SGDMRB

The frictional torque induced by the T-SGDMRB is based on two contributions from the magnetorheological fluid. The first is the torque ($T_V$) produced by the MR brake from the viscosity of the MR fluid in different channels without the effect of the magnetic field. The second torque ($T_M$) arises from the effect of the magnetic field passing perpendicularly through the MR fluid in various channels, causing a change in the MR fluid's properties at a high rate of viscosity. The total output braking torque ($T_b$) is the sum of these two induced torques from the different working modes of the MR fluid.

$$T_b = T_V + T_M \tag{1}$$

MR fluid controls the friction between the rotating disc and the stator. The state of MR fluid is addressed with the rate of the developed shear stress ($\tau$). Different models of MR fluid have been expressed by many researchers to describe shear stress. Rivanada et al. [22] predicted the shear stress of MR fluids using the Bingham plastic model, and the model is given in Equation (2) as follows.

$$\tau = \tau_M + \tau_V = \tau_M + \eta \frac{r\dot{\omega}}{d_p} \tag{2}$$

where $\tau_M$ and $\tau_V$ are the shear stress of MR fluid depending on the magnetic field and the shear stress independent of the magnetic field, respectively. $\eta$, $d_p$, and $\dot{\omega}$ stand for the MR fluid viscosity, channel gap contained by MR fluid, and signal from wheel rotation angular velocity, respectively. Briefly, shear stress ($\tau_V$) is a product of zero field viscosity of MR fluid and the magnitude of shear strain rate. The total torque generated by the MR brake in Equation (1) cannot be measured at the entire surface of the disc. The torque depends on the portion of surfaces in contact that are also under shear stress. However, torque upon a discretized small element is expressed as follows.

$$dT_b = \tau r dA = \tau r^2 dr d\theta \tag{3}$$

where $r$ is the radius from the center of rotation to the cross-section in shear that is represented by $dA$, considering a cartesian coordinate. $\theta$ is the spin angle from 0 to $2\pi$ defining the entire circumference of the MR fluid that is in contact. The total torque generated across the whole surface under shear stress is a double integral of Equation (3) with the radius and entire circumference bounds.

$$T_b = \int_r \int_\theta \tau r^2 dr d\theta = \int_r \int_0^{2\pi} \tau r^2 dr d\theta = 2\pi \int_r \tau r^2 dr \tag{4}$$

By inserting Equation (2) into Equation (4) and equating it with Equation (1), we can generalize an expression of the total torque generated by the MR brake as follows.

$$T_b = 2\pi \int_r r^2(\tau_M + \tau_V)dr = T_M + T_V$$
$$T_M = 2\pi \int r^2\tau_M dr, \ T_V = 2\pi \int r^2\tau_V dr$$

(5)

The proposed MR brake has a disc with a T-shape as the first consideration approach in our design configuration. Secondly, the disc has a groove in the middle of the outer surface. According to the design, the configuration includes those two approaches, and the total braking torque ($T_b$) is determined by four torque components with three subcomponents of each one, as expressed in Equation (6).

$$T_{MR_1} = 2\pi \int_{r_0}^{r_a} r^2\tau_{MR}dr = \frac{2\pi}{3}\tau_{MR}(r_a^3 - r_0^3), \ T_{MA_1} = 2\pi \int r_a^2\tau_{MA}dt_a = 2\pi \ \tau_{MA}r_a^2 t_a$$
$$T_{VR_1} = \frac{2\pi\eta}{d_p} \int_{r_0}^{r_a} r^3\dot{\omega}dr = \frac{\pi\eta}{2d_p}\dot{\omega}(r_a^4 - r_0^4), \ T_{VA_1} = \frac{2\pi\eta}{d_p} \int r_a^3\dot{\omega}dt_a = \frac{2\pi\eta}{d_p}r_a^3\dot{\omega}t_a$$
$$T_{MR_2} = 2\pi \int_{r_a}^{r_b} r^2\tau_{MR}dr = \frac{2\pi}{3}\tau_{MR}(r_b^3 - r_a^3), T_{MA_2} = 2\pi \int r_b^2\tau_{MA}dt_b = 2\pi \ \tau_{MA}r_b^2 t_b$$
$$T_{VR_2} = \frac{2\pi\eta}{d_p} \int_{r_a}^{r_b} r^3\dot{\omega}dr = \frac{\pi\eta}{2d_p}\dot{\omega}(r_b^4 - r_a^4), \quad T_{VA_2} = \frac{2\pi\eta}{d_p} \int r_b^3\dot{\omega}dt_b = \frac{2\pi\eta}{d_p}r_b^3\dot{\omega}t_b$$
$$T_{MR_3} = 2\pi \int_{r_c}^{r_b} r^2\tau_{MR}dr = \frac{2\pi}{3}\tau_{MR}(r_b^3 - r_c^3), T_{MA_3} = 2\pi \int r_c^2\tau_{MA}dt_c = 2\pi \ \tau_{MA}r_c^2 t_c$$
$$T_{VR_3} = \frac{2\pi\eta}{d_p} \int_{r_c}^{r_b} r^3\dot{\omega}dr = \frac{\pi\eta}{2d_p}\dot{\omega}(r_b^4 - r_c^4), \quad T_{VA_3} = \frac{2\pi\eta}{d_p} \int r_c^3\dot{\omega}dt_c = \frac{2\pi\eta}{d_p}r_c^3\dot{\omega}t_c$$

(6)

Therefore, the number of frictional surfaces of the disc is counted and multiplied by each calculated concerned individual torque. The total braking torque ($T_b$) output is a summation of $T_{MR}$, $T_{MA}$, $T_{VR}$ and $T_{VA}$. According to the states of the coil and the number of frictional surfaces of the disc, the final torque of the MR brake is expressed as follows.

$$T_M = 2T_{MR_1} + 2T_{MR2} + 2T_{MR3} + 2T_{MA1} + 2T_{MA2} + T_{MA3},$$
$$T_V = 2T_{VR_1} + 2T_{VR2} + 2T_{VR3} + 2T_{VA1} + 2T_{VA2} + T_{VA3}$$

(7)

The torque ($T_M$) is a final torque that includes different torques from radial and annular components. It is induced in the MR fluid to stop the rotation of the disc with the availability of a magnetic field when the coil is electrically supplied. Torque ($T_V$) includes all torques from both annular and radial paths of the MR fluid without electric current supplied to the coil. The insertion of Equation (7) into Equation (1) produces output torque measured from the T-SGDMRB, and it is expressed as follows.

$$T_b = T_M + T_V = 2T_{MR_1} + 2T_{MR2} + 2T_{MR3} + 2T_{MA1} + 2T_{MA2} + T_{MA3} + 2T_{VR_1}$$
$$+2T_{VR2} + 2T_{VR3} + 2T_{VA1} + 2T_{VA2} + T_{VA3}$$

(8)

### 2.3. Finite Element Simulation and Analysis of T-SGDMRB

The process of determining braking torque acquires the critical inputs from the MR fluid, coil, and surrounding magnetized and non-magnetized materials [23,24]. The magnetic materials create the magnetic pole of the magnetic flux lines. Non-magnetic materials make the direction of the magnetic flux lines change to improve the magnetic field from the coil [25]. Magnetic flux lines in the MR brake are distributed to pass freely through the magnetic materials [26]. The high-rate magnetization characteristic of steels selected for the disc and stator is possible when the change of magnetic field intensity is at 100 kA/m at its maximum, as shown in Figure 2a,b. The more the magnetic flux lines pass through the stator and disc, the more magnetic field is generated and arrives in the MR fluid channels.

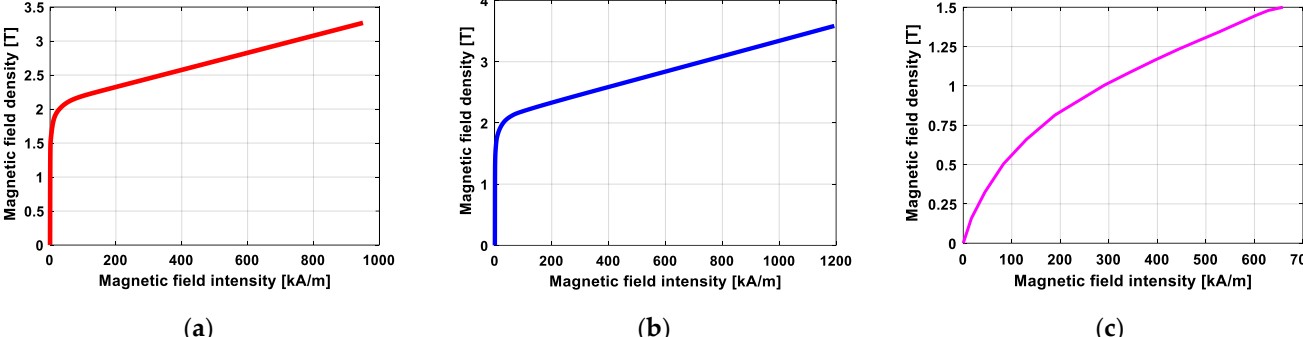

**Figure 2.** Magnetic properties of magnetized materials and MR fluid. (**a**) B–H curve of low carbon steel 1010 [27], (**b**) B–H curve of low carbon steel 1008 [28], (**c**) B–H curve of MRF- 132 DG [29].

During the material assigning of MR brake components in COMSOL, B–H curves of steels are used instead of the dimensionless relative permeability to determine the degree of magnetization passed through the stator and disc. Figure 2c shows the positive part of a B–H curve of MRF-132 DG from the technical datasheet. It allows the magnetic field to pass in the MR fluid before reaches its saturation value. The value of relative permeability is critically taken into consideration while defining material properties in COMSOL. A relative permeability of 4.5 was used in the simulation and calculated based on magnetic field density and intensity. The saturation magnetic flux density of MRF-132DG is 1.5 T. After magnetic parametrizing the model, the extremely fine meshes were generated to allow the further process of the finite element analysis.

Figure 3a shows the meshed model that contains a total number of 106, 351 mesh elements with $1.84 \times 10^{-8}$ of minimum element area, $3.49 \times 10^{-5}$ of maximum element area, and $3.315 \times 10^{-3}$ of the root mean square (RMS) edge length. Figure 3b shows the magnetic flux lines distribution along the magnetized materials and the MR fluid gaps. Magnetic flux lines that pass in the MR fluid are perpendicular to the flow of MR fluid in both the annular and radial paths. However, they are no longer strong at the same level, and the carried magnetic field densities are different. Figure 3c shows the magnetic flux density of the MR fluid with different currents. The maximum magnetic flux density is 0.53 T from the radial MR fluid gaps and 0.83 T from the annular MR fluid gaps. The quantity of the magnetic field in the MR fluid is not only dependent on the current supplied to the coil, but also on the number of magnetic flux lines that pass through the MR fluid and the distance between the MR fluid channel and the excitation coil [27]. The performance of MR fluid is examined by the amount of developed shear stress ($\tau$) once the excitation coil is electrically supplied. Shear stress in the MR fluid depends on the magnetic field ($\tau_M$) with the least squares method of the quartic polynomial function based on the magnetic field density of MR fluid [28]: it is expressed as follows.

$$\tau_M = C_4 B^4 + C_3 B^3 + C_2 B^2 + C_1 + C_0 \tag{9}$$

where $B$ is the magnetic flux density, $C_0, C_1, C_2, C_3, and\ C_4$ are polynomial fitting coefficients. $C_{0,1,2,3,4} = 0.1442$ Kpa, $13.708$ KpaT$^{-1}$, $158.79$ KpaT$^{-2}$, $-176.51$ KpaT$^{-3}$, and $52.962$ KpaT$^{-4}$. In this MR brake model, both ($\tau_M$) and ($\tau_V$) are involved in the scenarios of the parametric configuration of the T-SGDMRB. In this regard, the ($\tau_M$) is divided into ($\tau_{MR}$) and ($\tau_{MA}$) for the radial and annular MR fluid channels. Then, ($\tau_V$) is divided into ($\tau_{VR}$) and ($\tau_{VA}$) for radial and annular channels, respectively.

$$\tau_{MR} = C_4 B_R^4 + C_3 B_R^3 + C_2 B_R^2 + C_1 B_R + C_0$$
$$\tau_{MA} = C_4 B_A^4 + C_3 B_A^3 + C_2 B_A^2 + C_1 B_A + C_0 \tag{10}$$

where $B_R\ and\ B_A$ are the magnetic flux density in the MR fluid at the radial and annular channels. ($\tau_{VR}$) and ($\tau_{VA}$) are similar in their mathematical formulation because of constant

MR fluid gaps. It increases or decreases as the radius of circumference under viscous shear stress varies. Figure 4a,b shows the polynomial curve fitting for the developed yield stress ($\tau_M$) with magnetic field density (*B*) for both radial and annular channels. A high rate of yield stress of a value of 45.13 kPa is obtained in the annular channel ($\tau_{MA}$), and the maximum yield stress developed at the radial channel ($\tau_{MR}$) is 30 kPa. Figure 4c shows the generated torques from the different statuses of the coil. The viscous-generated torque in both the radial and annular channels is 5.26 Nm. When the coil is electrically supplied from 0 to 2.5 A, the generated torque in the annular channel reaches 127.04 Nm and 98.9 Nm in the radial channel. The total braking torque generated by this proposed model is 231.2 Nm.

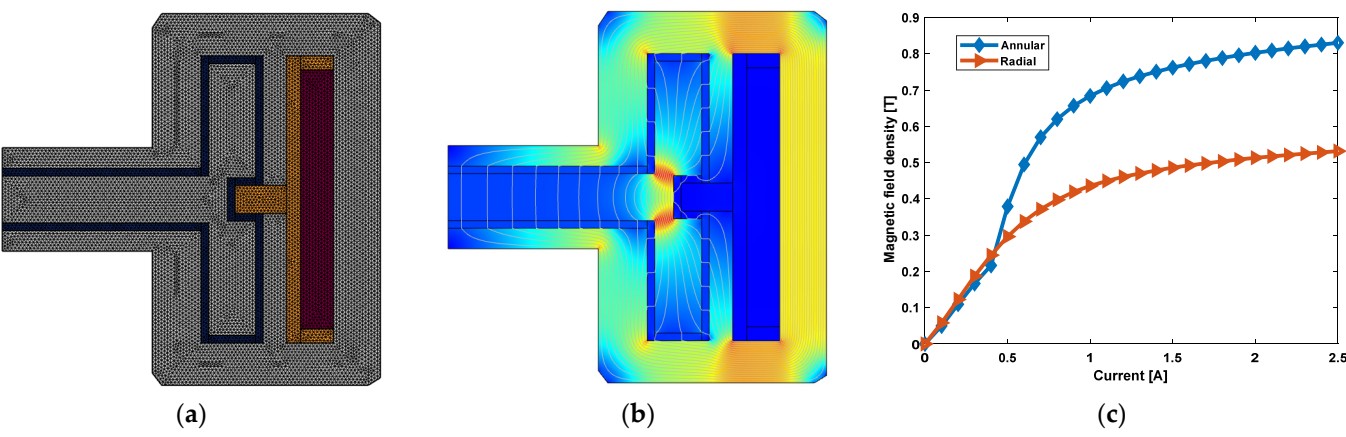

(**a**)      (**b**)      (**c**)

**Figure 3.** Finite element mesh and magnetic characteristics of the MR fluid after finite element simulation of the MR brake model. (**a**) Extremely fine meshed model, (**b**) Distributed magnetic flux lines, (**c**) Magnetic flux density with electric current.

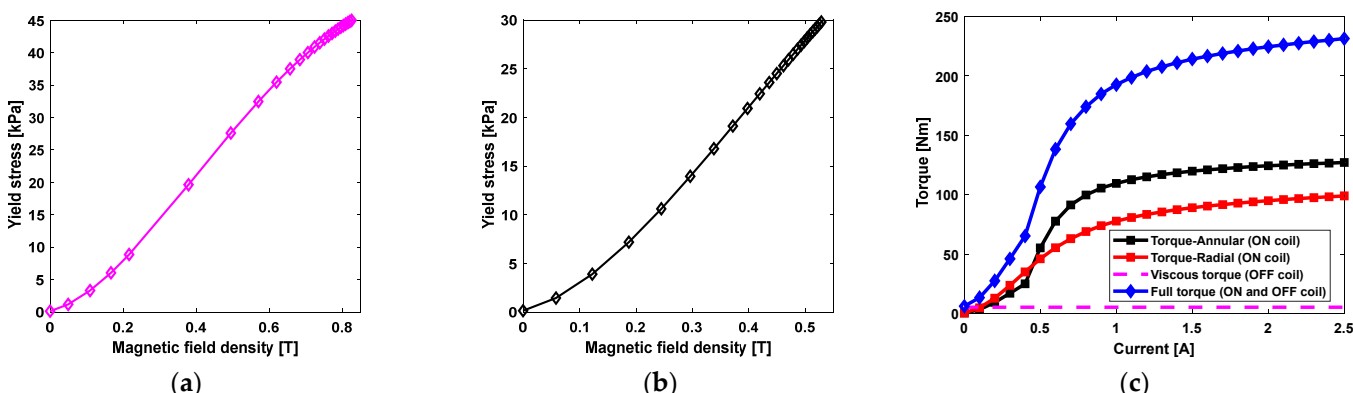

(**a**)      (**b**)      (**c**)

**Figure 4.** Designing results of the proposed model. (**a**) Yield stress with magnetic field density in the annular channel, (**b**) Yield stress with magnetic field density in the radial channel, (**c**) Generated braking torque with current.

### 2.4. Design Optimization of the T-SGDMRFB

The problem optimization of the proposed MR brake uses multi-objective optimization with the genetic algorithm (GA) method. Many complex design optimizations use this method because it uses mathematical formulations that include realistic models [29,30]. The purpose of this optimization is to determine the lightest T-SGDMRB that can fit in an available space and can produce the required braking torque to slow down or to stop a BMW R1200RT motorcycle at an operating temperature below the critical temperature of MRF-132 DG. The multi-objective optimization process of a T-SGDMRB is shown in Figure 5. Some constraints are made to parameterize the MR brake in the optimization process. Firstly, the disc hub has the same center as the motorcycle's wheel hub. It keeps being smaller with its low weight and is strong enough for torque transmission. It also has

fixed dimensions, and its mass does not vary but keeps correlating with other components in the optimality function. Secondly, the thickness of a grooved section keeps being smaller than the thickness of the disc leg to avoid bending the disc at high torque. Thirdly, $H_{h2} + h$ should be smaller than the inner radius of the motorcycle's wheel.

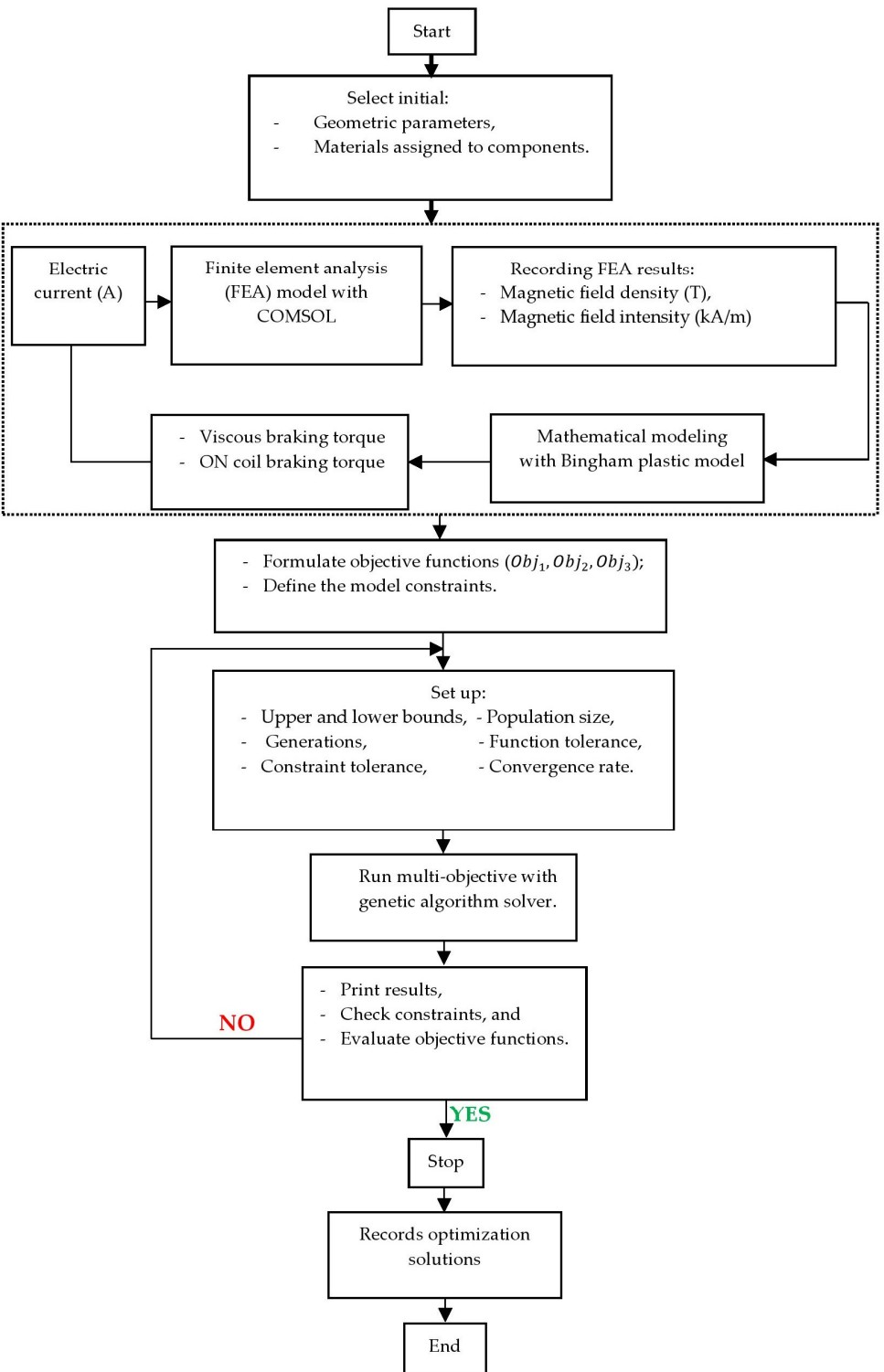

**Figure 5.** The flowchart of the process steps for multi-objective optimization with the genetic algorithm method. The steps include initial based inputs, parameterizing constraints, optimality function sets, processing approaches with the solver, and results decision of the optimal T-shape grooved disc MR brake.

The weight of the MR brake defines the cost and size of the MR brake. The mass is a function of material property ($\rho$) and geometric brake component volume ($V$). Then, the total mass of the T-SGDMRB is expressed in Equation (11).

$$m_b = V_{hb}\rho_{hb} + V_{st}\rho_{st} + V_{MRF}\rho_{MRF} + V_b\rho_b + V_c\rho_c + V_d\rho_d \tag{11}$$

where $V_{hb}$, $V_{st}$, $V_{MRF}$, $V_c$, and $V_d$ stand for the geometric volume of the disc hub, stator, MR fluid, bobbin, coil, and disc, respectively. $\rho_{hb}$, $\rho_{st}$, $\rho_{MRF}$, $\rho_b$, $\rho_c$, $\rho_d$ are density for the disc hub, stator, MR fluid, bobbin, coil, and disc. Their values are defined in Table 1.

**Table 1.** Material density of MR brake components.

| Component | Material | Density (kg/m$^3$) |
|---|---|---|
| Bobbin | Aluminum | 2700 |
| Coil | Copper | 8960 |
| Disc | Low carbon steel 1010 | 7870 |
| Disc hub | Stainless steel 316 | 7900 |
| MR fluid | MRF-132 DG | 2960 |
| Stator | Low carbon steel 1008 | 7870 |

The objective function ($Obj_1$) formed by the mass ($m_b$) minimizes the total weight of the MR brake in line with the available space reserved for the brake and reference mass ($m_{ref}$).

$$Obj_1 = g_1 \frac{1}{m_{ref}}(m_{ref} - m_b)$$
$$Subjected\ to: \begin{cases} m_{ref} > m_b \\ x_i^{U_b} \geq x_i^{L_b} \end{cases} \tag{12}$$

where $g_1$ represents a scalar factor to parametrize the mass of this MR brake for the good optimal individual variable, $x_i$ stands for the design variables ($i = 1, 2, 3, \ldots n\ variable$), $U_b$ and $L_b$ are upper and lower bounds, respectively.

With dynamic consideration of the wheel under braking, the resistance friction forces of the vehicle that can ideally represent braking force ($F_b$) applied on the wheel are equal to the longitudinal force ($F_l$). Braking force ($F_b$) is the ratio of a required generated braking torque ($T_b$) and the motorcycle's wheel radius ($w_R$), as can be stated as follows.

$$F_l = F_b = \frac{T_b}{w_R} \rightarrow T_b = m \cdot a_d \cdot w_R \tag{13}$$

where $m$ is the total mass of the motorcycle, and $a_d$ is a longitudinal braking deceleration. The maximum longitudinal braking deceleration ($a_d$) of a motorcycle is 0.75 g (g represents gravitational acceleration) without wheel slip and other road characteristics [31]. The wheel radius ($w_R$) is 267 mm and the total mass ($m$) is 495 kg (permissible gross weight of the motorcycle) from the specifications datasheet of the BMW R1200RT [32]. There is mass transfer to the front and rear wheels during braking, but this does not occur equally [33]. However, the braking torque ($T_b$) calculated in Equation (13) will be used as a reference torque ($T_{ref}$). The required braking torque for the wheels on the front axle is different from the one required to brake the wheels on the rear axle [34]. This braking principle is also applied to two-wheeled vehicles like motorcycles [35]. Mass transfer is a function of the total mass of the motorcycle and the motorcycle's dynamic characteristics, as expressed in Equation (14) [35]. The transferred mass to the wheels is larger on the front wheel than on the rear wheel. This is the reason for the different required braking torque applied to the two wheels of a motorcycle.

$$m_F = \frac{1}{L \cdot g}((m \cdot g \cdot L_R) - (m \cdot a_d \cdot h))$$
$$m_R = \frac{1}{L \cdot g}((m \cdot g \cdot L_F) - (m \cdot a_d \cdot h)) \tag{14}$$

where $m_F$ *and* $m_R$ are the motorcycle's mass transferred to the front wheel and the rear wheel. $h, L, L_F, L_R$ are motorcycle dynamic characteristics, which are listed in Table 2. From Equation (14), the total required braking torque ($T_{ref}$) for the motorcycle is 972.4 Nm. By inserting the individual front and rear masses from Equation (14) into Equation (13), the required braking torque for the front wheel is found to be 589.2 Nm, and the braking torque on the rear wheel is supposed to be 383.2 Nm. Referring to previous work and the properties of the selected MR fluid, it is evident that the proposed MR brake cannot produce the torque required for the front wheel. Therefore, to satisfy the wheel dynamics during braking, two MR brakes will be used on the front wheel, with each MR brake needing to produce at least 294.6 Nm of braking torque. Therefore, the objective function ($Obj_2$) is estimated by Equation (15).

$$Obj_2 = g_2 \frac{1}{T_{ref}}(T_b - T_{ref})$$
$$Subjected\ to: \begin{cases} T_{ref} < T_b \\ x_i^{U_b} \geq x_i^{L_b} \end{cases} \tag{15}$$

where $g_2$ is a scalar weight factor for braking torque to obtain an important correlation of the variables.

**Table 2.** Motorcycle dynamic characteristics.

| Input Parameter | Value |
| --- | --- |
| Wheel base ($L$) | 1507 (mm) |
| Horizontal position from the center of gravity to the front wheel ($L_F$) | 878.9 (mm) |
| Horizontal position from the center of gravity to the rear wheel ($L_R$) | 628.1 (mm) |
| Height to the center of gravity ($h$) | 380 (mm) |
| Wheel radius ($w_R$) | 267 (mm) |

The output braking torque is obtained from the generated shear rate between the MR fluid and its surrounding parts. During the operation of the MR brake, heat is generated in the MR fluid channels. The MR brake components play a role in cooling through conduction. However, the performance of the MR fluid can decrease when the MR brake operates above its allowable working temperature [36]. To avoid failure of the braking process, the operating temperature of the MR fluid should be within the prescribed range. The highest operating or cruising temperature should not exceed the maximum critical temperature of the MRF-132 DG (403.15 K) designated by the manufacturer, as expressed in Equation (16).

$$T_{MRF} \leq T_m - dT \tag{16}$$

where $T_{MRF}$, $T_m$, $dT$ are cruising temperature of MR fluid flows in channels, the maximum allowable working temperature of MR fluid, and the temperature increment during the braking process, respectively. In most cases, the incremental braking temperature is around 283.15 K and depends on the braking energy. It increases as the brake activates. Mathematically, dT is expressed as follows.

$$dT = \frac{\Delta E_b}{m_b \cdot C} = \frac{1}{m_b \cdot C}\left(\frac{m_{F/R}}{2} \cdot \frac{m \cdot V_i^2}{2}\right) \tag{17}$$

where $\Delta E_b$, $C$, *and* $V_i$ are the braking energy, brake equivalent specific heat, and cruising speed of the motorcycle, respectively. The braking torque measured from zero magnetic fields ($T_V$) is a result of converting thermal energy from the friction energy of MR fluid and the surrounding parts into heat. The steady cruising temperature of MR fluid ($T_{MRF}$) generated during this process is released into the environment using a heat transfer approach [37].

According to Newton's law of cooling, the rate of heat transfers out from the MR brake ($\dot{Q}_b$) is mathematically stated as follows.

$$\dot{Q}_b = h_c \cdot A_0 (T_{MRF}(t) - T_A) = T_V\left(\frac{V_i}{w_R}\right)$$
$$T_{MRF}(t) = \frac{1}{h_c \cdot A_0}\left(T_V\left(\frac{V_i}{w_R}\right)\right) + T_A \tag{18}$$

where $h_c$ is the heat transfer coefficient, and it is around 75 $W{\cdot}m^{-2}{\cdot}K^{-1}$; $A_0$ is a total heat transferred outer surface of the MR brake; and $T_A$ is an ambient temperature, which is settled to 303.15 $^0K$. The third objective function ($Obj_3$) is formed based on the cruising temperature of MR fluid, and it is expressed as follows.

$$Obj_3 = g_3 \frac{1}{T_m}(T_m - T_{MRF})$$
$$Subjected\ to: \begin{cases} T_m > T_{MRF} \\ x_i^{U_b} \geq x_i^{L_b} \end{cases} \tag{19}$$

where $g_3$ is a scalar weight factor of MR fluid temperature. The scalar factors are adjusted from 0 to 1 by checking the correlation of variables. The next step of optimization is to solve the problem by evaluating the formed objective functions or fitness values. The multi-objective optimization algorithm is carried out in MATLAB software. It starts by evaluating the fitness values of each generation; then, the best individuals among them are chosen for the next population. This process continues until the objective functions reach their convergence limits, which is the end of this optimization process.

Figure 6a,b shows the Pareto curves with full attention to the settled objective functions assigned to the proposed MR brake. Braking torque increases with the increase in mass. Torque increases slowly before 5 kg, when mass increases significantly. After 5.5 kg of MR brake mass, the mass increases slowly while the braking torque increases at a higher rate until the objective function converges. The braking torque increases as the operating cruising temperature increases. The torque increases continuously with temperature until 297.1 Nm, when the cruising temperature is 86.7 °C. Within a temperature range of 86.7 °C and 87.4 °C, the braking torque remains constant and decreases with 0.2 Nm. After 87.4 °C, the braking torque keeps increasing with cruising temperature, but the temperature increases slowly. The braking torque is the main function, and other parameters vary to impact the torque to reach its reference value. Figure 6c,d shows the design variables after deciding the convergence of the fitness functions. Design variables with the same center as the disc hub impact the brake diameter and torques on the radial channel. The other design variables impact the thickness of the MR brake, and their variation to converge affects the annular braking torques. All design variables are converged at the 50th generation.

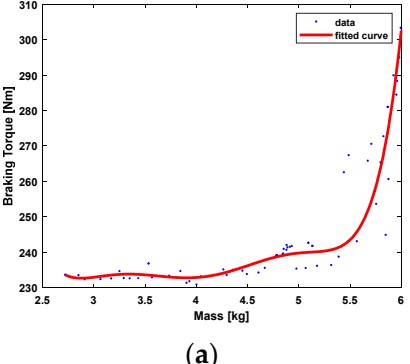

(a)

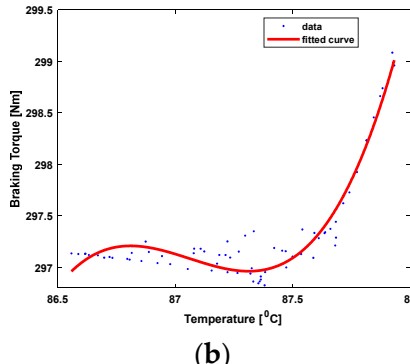

(b)

**Figure 6.** *Cont.*

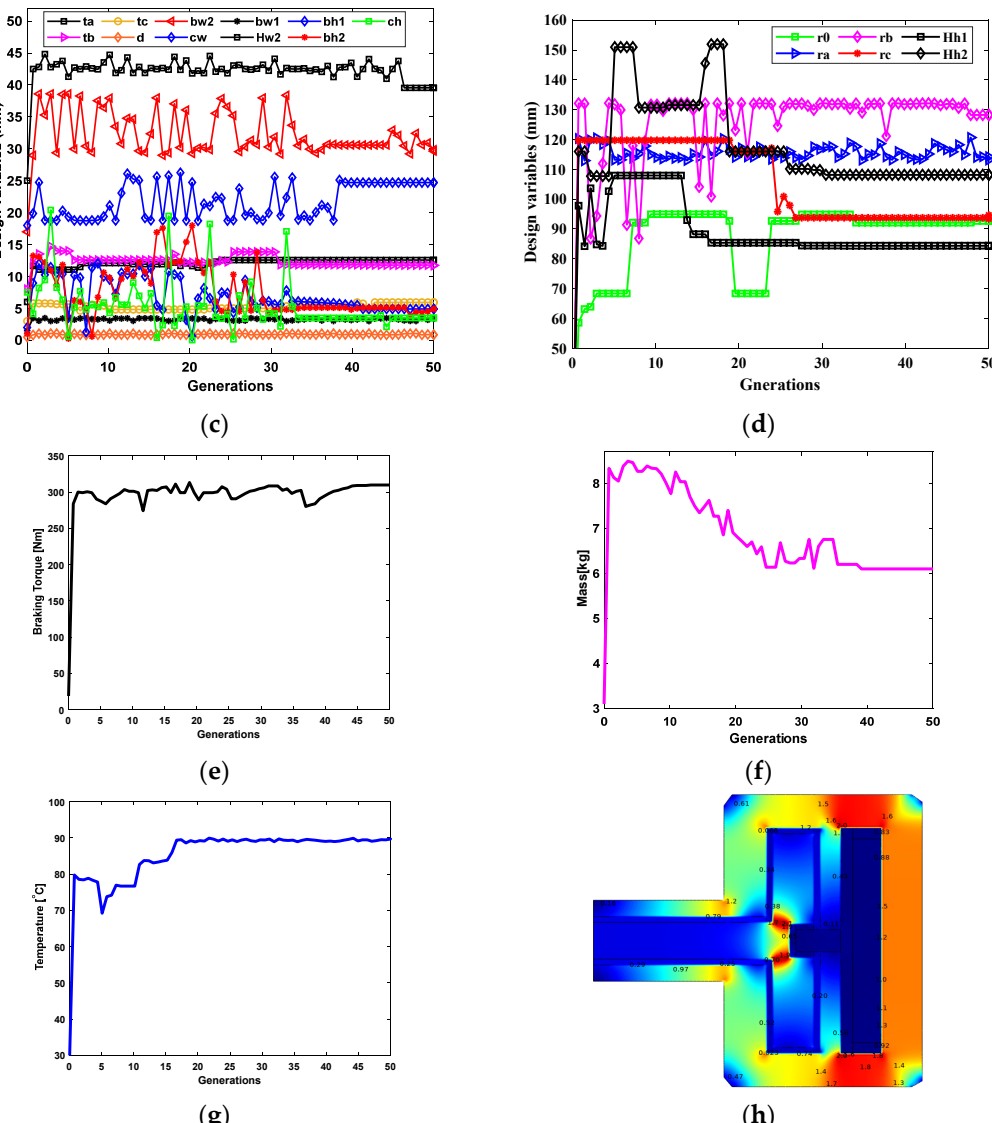

**Figure 6.** Optimization results of T-shape grooved disc MR brake. (**a**) Pareto front between objective functions ($Obj_1$ and $Obj_2$) for the mass of MR brake and torque, (**b**) Pareto front between objective functions ($Obj_2$ and $Obj_3$) for torque and cruising temperature, (**c**) Design variables that vary with MR brake diameter, (**d**) Design variables that vary with MR brake thickness, (**e**) MR braking torque with generations, (**f**) Mass of MR brake with generations, (**g**) Cruising temperature with generations, (**h**) Optimized magnetic field density along the magnetic materials.

Figure 6e shows the achieved braking torque curve at the converged fitness functions. The maximum converged value of the optimal braking torque is 303.9 Nm but is achieved after the 45th generation. Figure 6f shows the optimal mass of the MR brake when its fitness function has finally converged. The optimal weight of the T-SGDMRB is 6.1 Kg. Figure 6g shows the optimal curve of the operating cruising temperature after the convergence of the fitness function. The temperature increases unsteadily and keeps the maximum of nearly 90 °C, which is below the critical saturation temperature of MRF-132 DG. Figure 6h shows the optimized magnetic flux density distribution among the magnetized materials. The magnetic flux density is higher in the stator near the excitation coil and the part of a disc between the disc leg and disc flange at the grooved part. The lowest magnetic field densities are also found on the disc and housing near the disc hub. The global optimal vector (*x*) results and the initial dimensions are listed in Table 3. The initial dimensions of each brake component were correlated with the components' geometric structure and fitness functions.

During the optimization process, the geometric structures of brake components do not change but the dimensions have been changed to reach the reference inputs of the objective functions. The optimal dimensions have been recorded after the 50th generation and are the geometric dimensions for an optimally designed MR brake.

**Table 3.** Optimum solutions of geometric parameters for T-shape grooved disc MR brake.

| MR Brake Component | Geometric Parameter | Dimension (mm) Initial | Optimal | MR Brake Component | Geometric Parameter | Dimension (mm) Initial | Optimal |
|---|---|---|---|---|---|---|---|
| Disc hub | Inner radius ($r$) | 37 | 37 | Stator | Fillet length($h$) | 2 | 2 |
| | Thickness ($t_h$) | 4 | 4 | | Inner radius ($H_{h1}$) | 83 | 108 |
| | Outer radius ($r_1$) | 47 | 47 | | Outer radius ($H_{h2}$) | 136.5 | 133 |
| | Outer radius ($r_0$) | 69 | 92 | | Thickness ($H_{w2}$) | 41 | 36.6 |
| Disc flange | Inner radius ($r_a$) | 90 | 113.2 | Bobbin | Length ($b_{h1}$) | 4.5 | 5.5 |
| | Outer radius ($r_b$) | 100 | 120.8 | | Width ($b_{w1}$) | 2 | 3 |
| | Bottom width2($t_a$) | 10 | 12 | | Length ($b_{h2}$) | 5.5 | 5 |
| | Upper width 2($t_b$) | 12 | 12.2 | | Width ($b_{w2}$) | 30 | 30.6 |
| | Thickness ($t_k$) | 6 | 5 | | | | |
| Disc leg | Inner radius ($r_o$) | 69 | 92 | Coil | Length ($C_h$) | 3 | 4.5 |
| | Outer radius ($r_a$) | 90 | 113.2 | | Width ($C_w$) | 24 | 27.5 |
| | Thickness ($t_k$) | 6 | 5 | | Number of turns | 175 | 300 |
| Groove | Radius ($r_c$) | 96 | 116 | MR fluid gap | Annular gaps ($d$) | 1.5 | 0.8 |
| | Thickness ($t_c$) | 5 | 4.6 | | Radial gaps ($d$) | 1 | 0.8 |

Thereafter, optimal dimensions of design variables and magnetic field density are used to recalculate the optimal yield stress and braking torque of a T-SGDMRB. Figure 7a,b shows the yield stress developed in the optimized MR fluid. They describe the performance of the MR fluid with an applied magnetic field for the optimal MR brake model. The highest developed yield stress for our model is 36.56 kPa from an annular channel, which is below the critical saturation yield stress of this MR fluid. The maximum yield stress in the radial channel ($\tau_{MR}$) is 23.86 kPa. The saturation yield stress from the technical datasheet of Lord Corporation is 50 kPa [38,39].

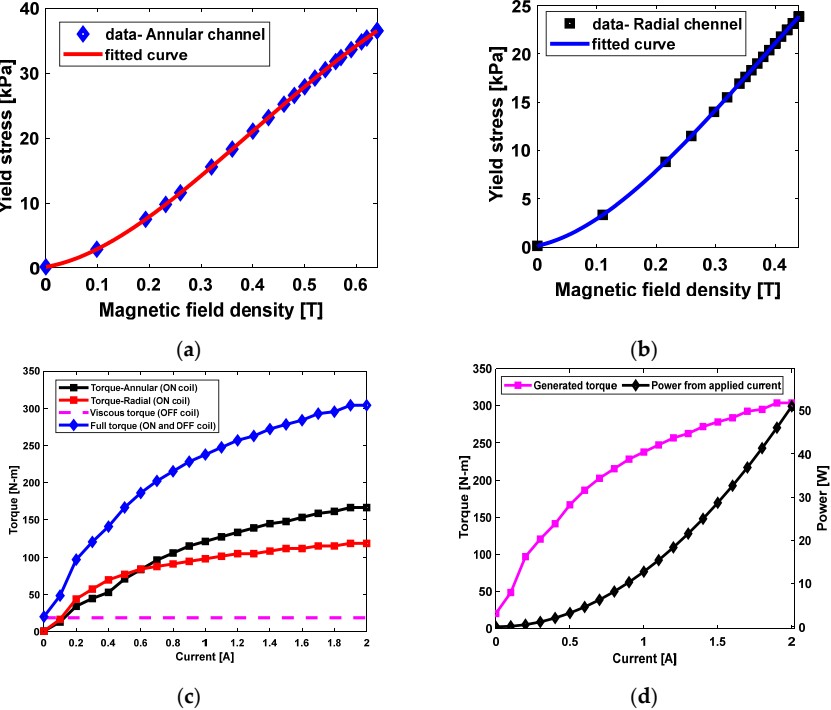

**Figure 7.** Designing results from an optimal model. (**a**) Polynomial curve fitting for the yield stress ($\tau_{MA} - B_A$) in the annular channel (**b**) Polynomial curve fitting for the yield stress ($\tau_{MR} - B_R$) curve in the radial channel, (**c**) Generated braking torque output, (**d**) Electric power consumption with generated torque.

Figure 7c shows the generated braking torque from the optimally designed MR brake model. The yield stress from the simulation and calculus is inserted into the torque equations for the torque analysis concerning the Bingham plastic model. Braking torque increases as the supplied electric current on the excitation coil increases. When the excitation coil is electrically zeroed, the viscous torque from both annular and radial channels reaches 18.72 Nm. With the coil supplying 0.1 to 2 A, the torque measured in the radial channel of MR fluid is 118.55 Nm; and the torque measured in the annular channel is 166.69 Nm. The total generated braking torque is 303.9 Nm, resulting from the summation of radial, annular, and viscous torques. However, the reference braking torque (294.6 Nm) of a single MR brake for the front wheel is only achieved with 1.8 A. From the supply current of 2 A, the electric power consumption of the coil is 51 W to achieve the maximum torque of 303.9 Nm, as shown in Figure 7d. The consumed power is proportional to the current and the coil resistance when considering the total length of the required wire [26]. The required torque of the rear wheel will be achieved after parameterizing the controller based on this maximum generated torque.

## 3. Control Performance Evaluation of T-SGDMRB

### 3.1. System Identification of Brake Actuation and Wheel Slip Control

In this section, a designed MR brake is assembled on the motorcycle "BMW R1200RT" through mathematical algorithms and MATLAB Simulink to evaluate the braking phenomena. Both in the actual mode and simulation mode, the braking system is used to slow down the vehicle or keep the vehicle stationary [40,41]. Motorcycle dynamics, road characteristics, and controlled braking torque are used to formulate different mathematical models. Some required parameters of the motorcycle in mathematical modeling and simulation are shown in Figure 8, and their values are listed in Table 4. In the simulation, an extended Kalman filter (EKF) controller is chosen to deal with such kinds of inputs.

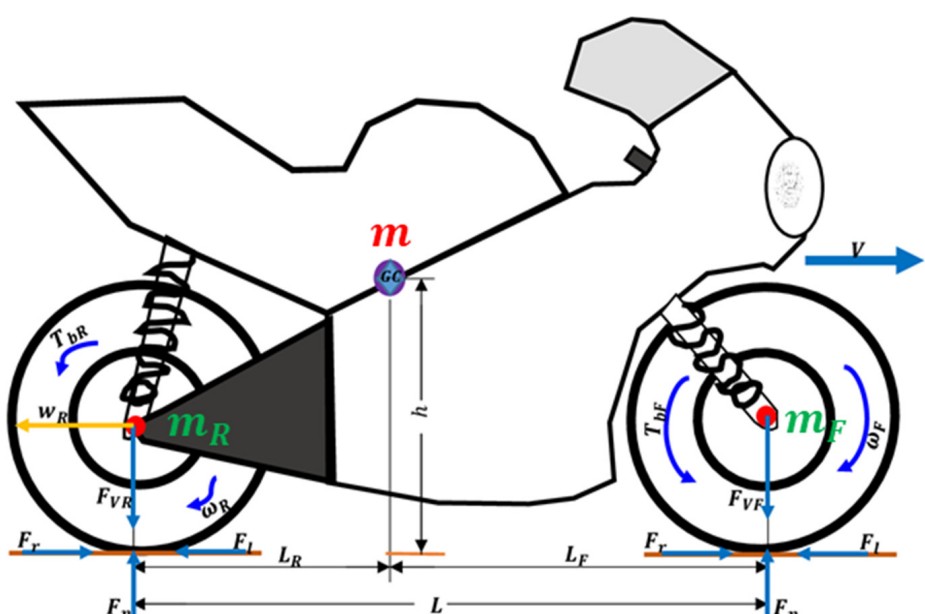

**Figure 8.** Motorcycle and wheels dynamic layout.

An extended Kalman filter is a significant controller that linearizes the nonlinear process, measurement, and processing noises in the system for the estimation of the current state [42]. The extended Kalman filter controller competes to control slip in the rear and front wheels of the motorcycle to improve driving safety in skidding conditions. The simulation used to control slip is based on mathematical models of road conditions applied to a controlled model to display both actual and estimated signals of linear velocity, wheel slip rate, and wheel angular velocity with respect to the feed-forward path of the control

loop, as shown in Figure 9. It is complicated to control the slip of the rear and front wheels for the motorcycle simultaneously, but it improves motorcycle stability, braking distance, and braking time [43].

**Table 4.** Parameters of motorcycle.

| Input Parameter | Value |
|---|---|
| Permissible gross weight (m) | 495 (kg) |
| Wheel radius ($w_R$) | 0.267 (m) |
| Motorcycle front cross-sectional area ($A_f$) | 0.82 (m$^2$) |
| Air density ($\rho_a$) | 1 (kg/m$^3$) |
| Drag coefficient ($C_{df}$) | 0.75 |
| Mass of the wheel ($m_w$) | 6.08 kg |
| Initial braking speed ($V_i$) | 180 (kw/h) |

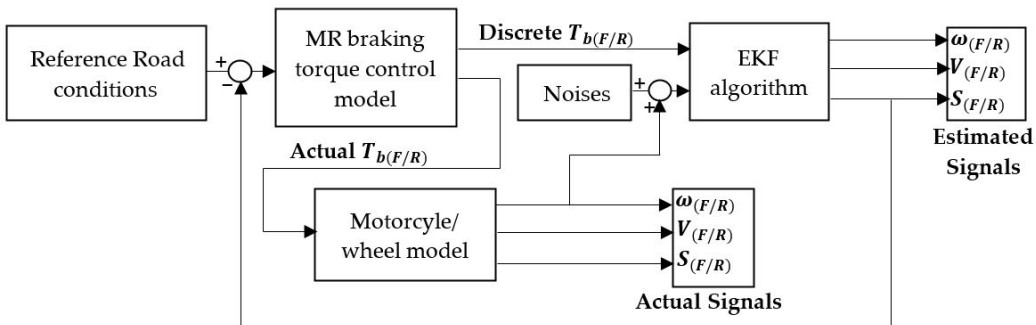

**Figure 9.** Structure of wheel slip control loop using the extended Kalman filter.

*3.2. Road Characteristic*

The road surface defines the contact patch between the motorcycle wheel and the road. The forces applied to the wheel and the friction coefficient (C) are all required to rotate the wheel [44,45]. The friction coefficient between the wheel and the road depends on different factors, including wheel skid (S), longitudinal motorcycle speed (V), type of road surface, and environmental conditions like humidity and high temperature [46]. To predict the vehicle longitudinal speed and wheel velocity at the actual and estimated output during simulation, the reference slip rates of both the front and rear wheels ($S_{(F/R)}$) are set to one. The wheel slip rate in mathematical and simulation models is the ratio of the velocity of the wheel ($\omega_{(F/R)}$) and an equivalent speed of a motorcycle (V).

$$S_{(F/R)} = 1 - \frac{\omega_{(F/R)} w_R}{V} \tag{20}$$

The improved magic formula model by Joan et al. [47] is adopted to calculate the wheel–road friction coefficient (C). It is a function of the reference slip rate of a wheel on a certain road in percentage ($S_r$) and the road coefficients that are represented by ($\xi_{1,2,3}$). Some typical road characteristics ($\xi$ and $S_r$) are shown in Table 5.

**Table 5.** Road types with characteristic parameters.

| Type of Road | $\xi_1$ | $\xi_2$ | $\xi_3$ | $S_r$ |
|---|---|---|---|---|
| Wet asphalt | 0.857 | 33.82 | 0.35 | 0.80 |
| Dry asphalt | 1.280 | 23.99 | 0.52 | 1.17 |
| Wet cobble | 0.400 | 33.71 | 0.12 | 0.38 |
| Dry cobble | 1.371 | 6.46 | 0.67 | 1 |
| Dry cement | 1.197 | 25.17 | 0.54 | 1.09 |
| Snow | 0.195 | 94 | 0.065 | 0.14 |

The coefficient of friction alone cannot describe the best road for simulation but can indicate the length of the stability zone and instability zone of the road. Other factors including longitudinal deceleration ($a_d$) and braking balance ($b_l$) should also be taken into consideration. These factors are expressed in Equation (21).

$$C = \xi_1(1 - e^{(-\xi_2 S_r)} - \xi_3 S_r)$$
$$a_d = Cg, \; b_l = \frac{(m\,g\,L_R)}{L} + \frac{a_d m\,h}{L} + d_f / \frac{(m\,g\,L_F)}{L} - \frac{a_d m\,h}{L} + d_f \qquad (21)$$

where $d_f$ is the aerodynamic drag force applied on a motorcycle. Figure 10 shows the friction coefficients and the slip rates for different roads. The dry asphalt road and dry cement reach large road friction coefficients at the minimum percentage slip rate. The other roads have very low friction coefficients in the adhesion region. The dry asphalt and dry cement roads are better for riding motorcycles compared to the other remaining roads in the studied scenarios. The snow road is the worst one to ride a motorcycle due to having its lowest friction coefficient in both adhesion and unstable regions.

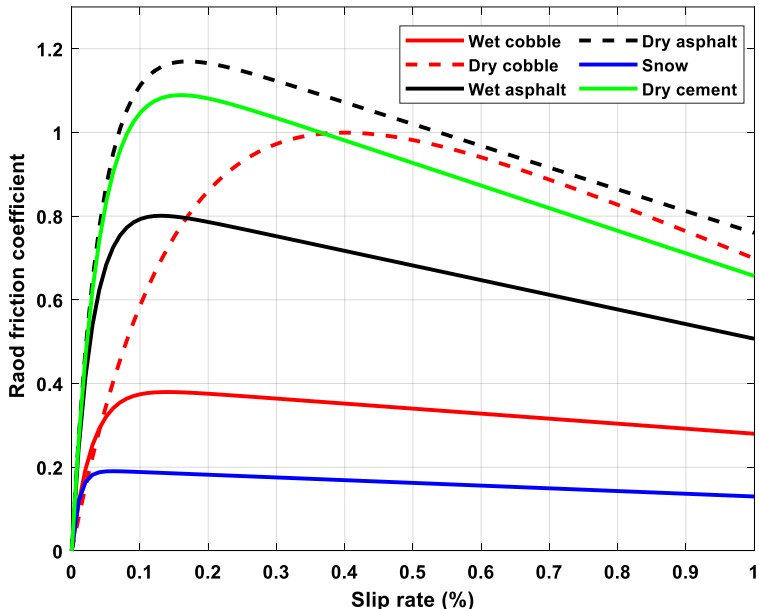

**Figure 10.** Typical roads and wheel contact characteristics.

The braking torque distribution required to brake the motorcycle is 55% on the front wheel and 45% on the rear wheel [1]. This distribution of braking torque leads to a braking balance of 1.2%. Figure 11 shows the braking balance with deceleration on different roads. The required braking balance is achieved at 4.7 $m/s^2$ of longitudinal deceleration from dry asphalt road while other roads need higher deceleration rates to achieve it. Therefore, the dry asphalt road is selected to be used in further analysis for evaluating the performance of the designed MR brake applied to a motorcycle (BMW R1200RT).

*3.3. MR Braking Torque with Dynamic Constraints*

The output braking torque from FEA and mathematical modeling is continuous over time. However, to reduce wheel lock-up or slipping, this braking torque needs to be discretized, becoming discontinuous over time. Figure 12 presents the block diagram of the MR braking torque discretization process. The generated torque signal is a function of road conditions and will serve as the direct input signal for the extended Kalman filter, which, in turn, affects the braking torque output applied to the wheel as a function of the feedback output signal ($\omega_{(F/R)}$) from the extended Kalman filter.

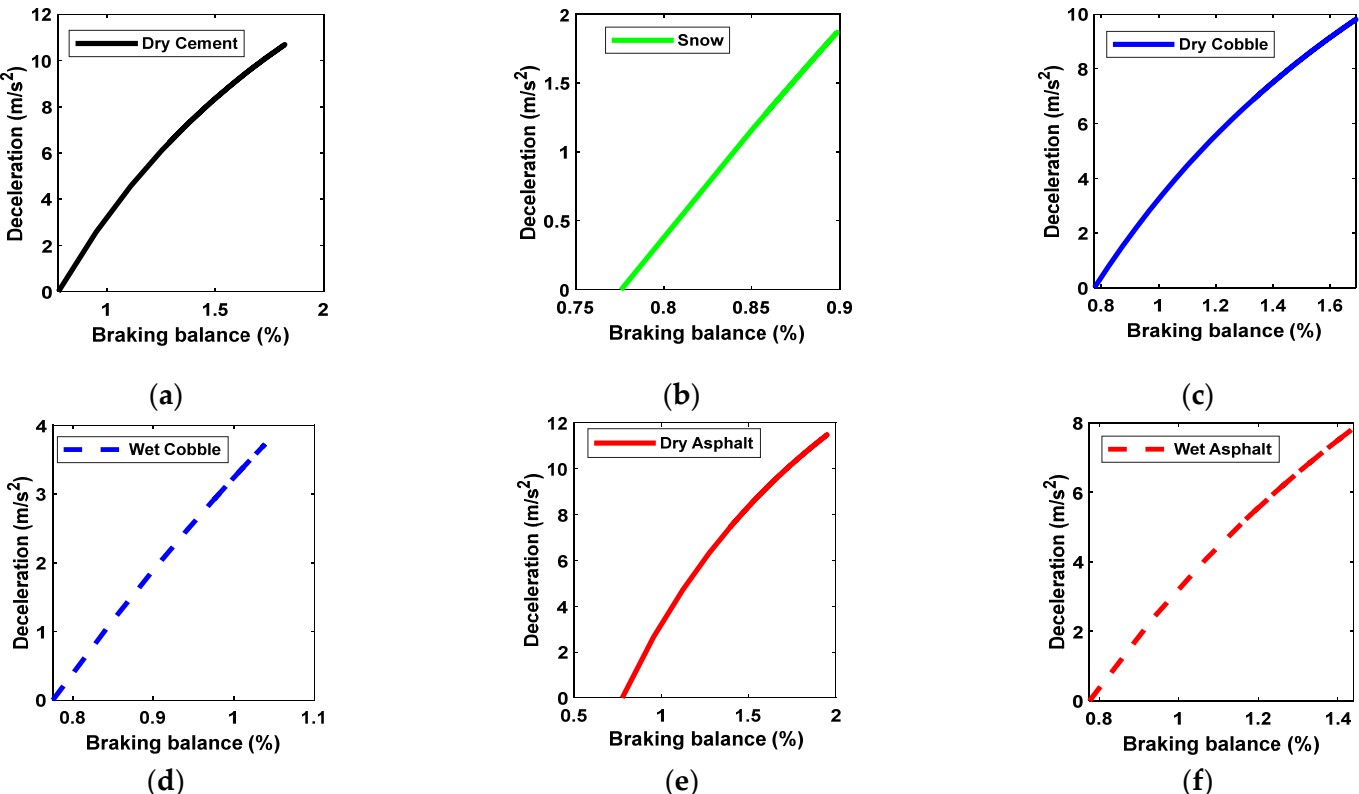

**Figure 11.** Deceleration with braking balance on different roads. (**a**) Dry cement, (**b**) Snow, (**c**) Dry Cobble, (**d**) Wet cobble, (**e**) Dry asphalt, (**f**) Wet asphalt.

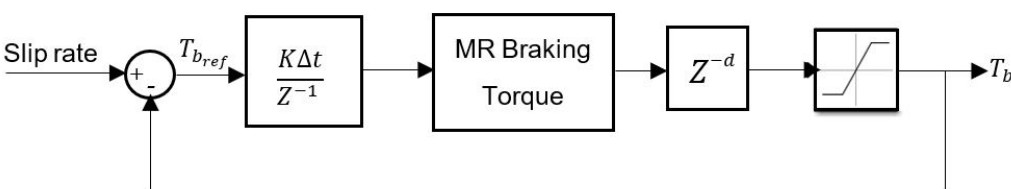

**Figure 12.** Closed loop block diagram of braking toque in wheel slip control.

### 3.4. Motorcycle Wheel Dynamic Model

The motorcycle's mathematical approach and simulation models include the real motorcycle with dynamic behaviors and wheel–road interactions. The rear and front wheels' braking actions are provided by optimal T-SGDMRB, as was discussed in the previous section. The MR brakes on motorcycle wheels are used to decelerate the motorcycle smoothly within a fast stopping time [48]. The different forces acting on each motorcycle's wheel during braking are rolling resistance force ($F_r$), longitudinal friction force ($F_l$), the normal force ($F_n$), and vertical weight ($F_v$), as shown in Figure 8. During braking, the transferred load applied to the rear wheel differs from the one on the front wheel, as described in Equation (15). The braking action is achieved when the wheel's angular velocity ($\omega_{(F/R)}$) is reduced to the possible minimum value. With Newton's second law of motion, the sum of all torques applied to the wheel center is zero.

$$T_{b(F/R)} + F_l \, w_R + J_0 \frac{d\omega_{(F/R)}}{dt} = 0$$

$$F_l = F_r = C \, F_{v(F/R)}, \quad F_{v(F/R)} = m_{(F/R)} \, g \text{ and}$$

$$F_l = m\frac{dV}{dt}, \quad \frac{dV}{dt} = \frac{1}{m}(C \, m \, g) = \frac{1}{m}F_l, \quad \frac{d\omega_{(F/R)}}{dt} = \frac{1}{J_0}\left(T_{b(F/R)} - F_l \, w_R\right) \tag{22}$$

where $J_0$ is the wheel's moment of inertia. The rear wheel and front wheel are equal in size and weight. The moment of inertia ($J_0$) is half of the product of the wheel mass and its squared radius.

### 3.5. Controller Modeling

The extended Kalman filter uses equations from road characteristics and wheel dynamic models to define its working process. It starts with the discretization of the continuous-time differential equations from Equations (20)–(22). EKF cannot trigger continuous-time functions or continuous signal inputs [49]. Therefore, those equations will be discretized based on a particular discrete period indexed ($k$), referred to as its previous point period that is indexed ($k-1$).

$$Z = \begin{Bmatrix} S_{(F/R)} \\ C \\ F_l \\ F_v \\ \frac{dV}{dt} \\ \frac{d\omega_{(F/R)}}{dt} \end{Bmatrix}, \; Z_k = \begin{Bmatrix} S_{(F/R)_k} \\ C_k \\ F_{l_k} \\ F_{v_k} \\ V_k \\ \omega_{(F/R)} \end{Bmatrix} = \begin{Bmatrix} \frac{V_{(k-1)} - w_R \omega_{(F/R)_{(k-1)}}}{V_{(k-1)}} \\ \xi_1 \left( \xi_2 e^{\left(-\xi_2 S_{(F/R)_{(k-1)}}\right)} - \xi_3 \right) \\ F_{v(F/R)_k} C_k \\ m_{(F/R)_k} g \\ V_{(k-1)} - \frac{\Delta t}{m} F_{l_{(k-1)}} \\ \omega_{(F/R)_{k-1}} + \frac{\Delta t}{J_0} \left( Sign\left(\omega_{(F/R)_{(k-1)}}\right) T_{b_{(F/R)}} - F_{l_{(k-1)}} w_R \right) \end{Bmatrix} \tag{23}$$

where $Z$ and $Z_k$ represent continuous and discretized states, respectively. The principal input of the extended Kalman filter is a discretized MR braking torque signal ($T_b$). The wheel's rotation velocity $\left(\omega_{(F/R)}\right)$ is considered a measurable variable entity and is also used as an input for the EKF, along with noise from roads and environmental conditions. Other variables are utilized as the state vector, all of which will be estimated by the EKF. The estimation state includes both the measured input ($Z_{kin}$) and the state vector ($\hat{Z}_k$).

$$Z_{kin} = \omega_{(F/R)}, \; \hat{Z}_k = \begin{Bmatrix} \hat{\omega}_{(F/R)_k} \\ \hat{V}_k \\ \hat{S}_{(F/R)_k} \\ \hat{C}_k \\ \hat{m}_k \end{Bmatrix} \tag{24}$$

The estimated variables are arbitrarily chosen, and two Jacobians ($H_k$), and ($P_k$) will be calculated using the discrete system equations with the measured input and estimated states to facilitate the predictor–corrector algorithm of the extended Kalman filter. The Jacobian matrices are expressed as follows.

$$H_k = \begin{bmatrix} 1 & 0 & 0 & \frac{\Delta t}{J_0}\left(w_R g \hat{m}_{(F/R)_k}\right) & \frac{\Delta t}{J_0}\left(w_R g \hat{C}_k\right) \\ 0 & 1 & 0 & -\Delta t g & 0 \\ -\frac{w_R}{\hat{V}_k} & \frac{w_R \hat{\omega}_{(F/R)_k}}{(\hat{V}_k)^2} & 0 & 0 & 0 \\ 0 & 0 & \xi_1\left(\xi_2 e^{-\xi_2 \hat{S}_{(F/R)_k}} - \xi_3\right) & 0 & 0 \\ 0 & 0 & 0 & 0 & 1 \end{bmatrix} \tag{25}$$

$$P_k = [1\,0\,0\,0\,0]$$

The mass described in the last row of the Jacobian matrix ($H_k$) seems to be a constant, but it varies slowly due to the stochastic nature of the extended Kalman filter. The errors in the estimated state are identified by the propagated covariance matrix ($Q_k$), which plays a

significant role in the Kalman filter gain ($G_k$). During control, EKF observation is evaluated at this step. Both are expressed as follows.

$$Q_k = H_k Q_k H_k^T + R,$$
$$G_k = \frac{Q_k P_k^T}{P_k Q_k P_k^T + M} \tag{26}$$

where ($R$) is the covariance of the process noise, and ($M$) is the covariance of the measured noise. The measurement errors here are approximately the same as the measurement noises and are calculated as the difference between the input measurements of the extended Kalman filter and the wheel speed $\left(\omega_{(F/R)}\right)$ from the motorcycle system model. Therefore, the summation of the $\left(\omega_{(F/R)}\right)$ and the product of errors with Kalman gain ($G_k$) provides the updated estimated state and covariance.

The control outputs of MR braking torque on the rear and front wheels are shown in Figure 13a. These braking torque responses are not constant over time due to the inputs from road characteristics and feedback signals of the estimated slip rate of wheels. However, they indicate that an MR braking torque of 589.2 Nm is applied to the front wheel, while the rear wheel experiences 383.2 Nm of braking torque. The braking torque responses return to their lowest value at the last second of the simulation because brake actuation should be released when the motorcycle's wheel reaches its lowest possible angular velocity or comes to a complete stop.

Figure 13b shows the estimated slip rate of the rear wheel, which significantly overshoots beyond the maximum optimal slip rate of the dry asphalt road. After the 4th second, the slip rate of the rear wheel reaches 100%. In Figure 13c, the estimated slip rate of the front wheel also shows overshooting above the optimal slip rate, though it is not as high as that of the rear wheel. Both wheels were controlled using the same measured noise input of the EKF. The optimal slip rate of the dry asphalt road is 0.17, which was the control target of our method. Slip rate errors (measured residual slip rates) of both wheels are shown in Figure 13d,e. These slip rate errors are removed from the slip rates of the estimated states to achieve the target slip rate.

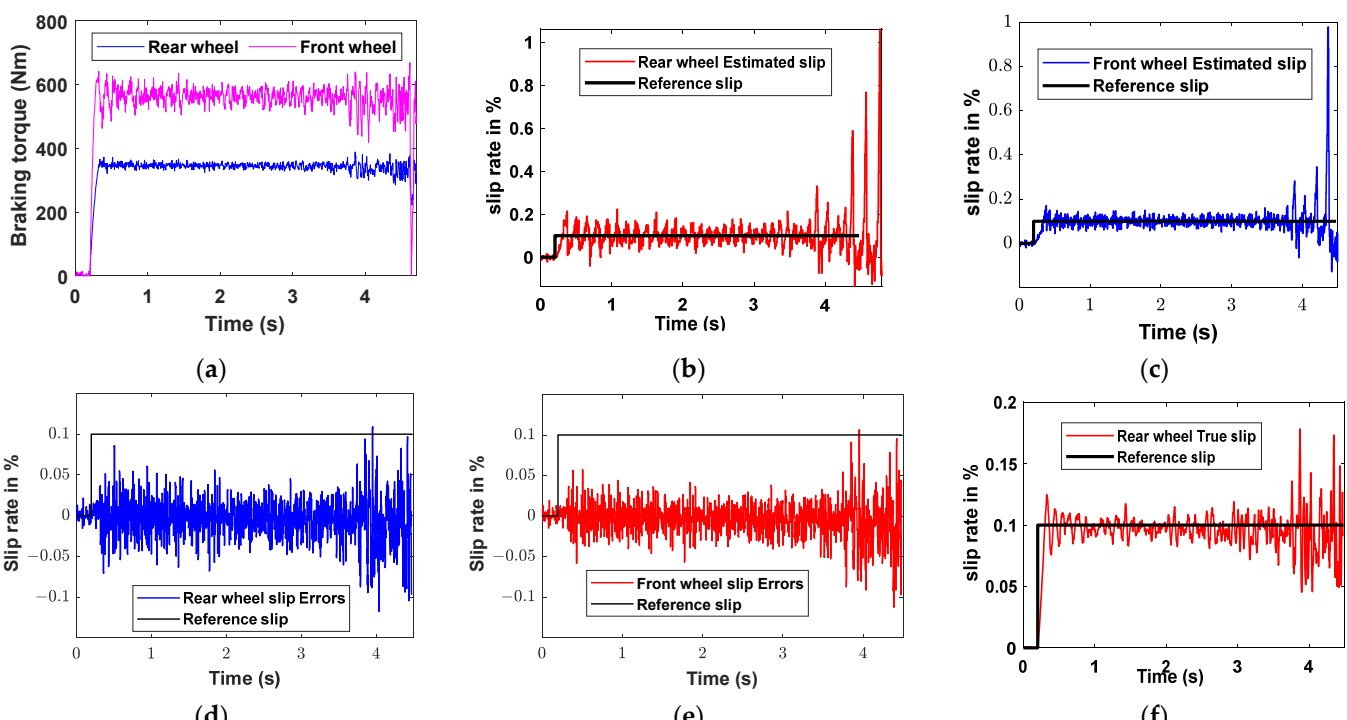

**Figure 13.** *Cont*.

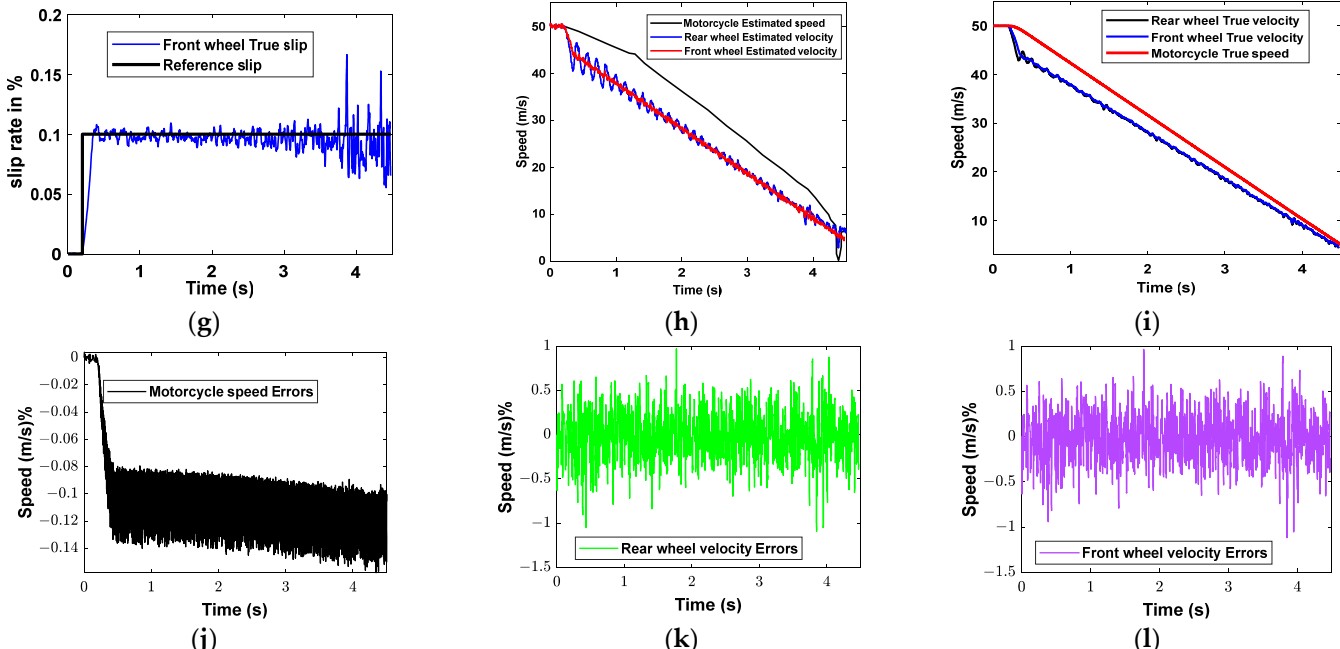

**Figure 13.** Simulation results. (**a**) Rear and front wheel MR braking torque responses, (**b**) Estimated slip rate of the rear wheel, (**c**) Estimated slip rate of the front wheel, (**d**) Slip rate error rear wheel, (**e**) Slip rate error front wheel, (**f**) Actual slip rate for the rear wheel, (**g**) Actual slip rate of the front wheel, (**h**) Estimated speeds, (**i**) Actual speeds, (**j**) Motorcycle speed errors, (**k**) Rear wheel velocity errors, (**l**) Front wheel velocity errors.

Figure 13f shows the true slip rate responses of the rear wheel, which are closer to the reference slip since they start at 3.87 s. After this time, the slip rate responses increase, but not above the maximum allowable optimal slip rate. The front wheel presents better slip rate responses compared to the rear wheel, as shown in Figure 13g. The true slip rate responses of the front wheel are very close to the reference slip, and its highest overshoot is 0.15, measured at 4.3 s.

Figure 13h shows the estimated speeds of the motorcycle and its wheels under braking. The longitudinal speed of the motorcycle decreases along with the wheels' velocities, but not smoothly. The rear wheel's velocity curve presents more lock-up compared to the front wheel. After 3 s, the speeds of the motorcycle and both wheels coincide due to an increase in the wheels' slipping rates.

Figure 13i shows the actual longitudinal speed of the motorcycle and the wheels' velocities. Speeds are decreasing smoothly compared to those in the estimated state. The longitudinal speed of the motorcycle decreases from 50 m/s until it stops at the end of the simulation without any up-or-down concavity. The rear wheel's velocity curve is more oscillating than that of the front wheel, but both look smoother compared to the wheel velocity curves from the estimated state.

Figure 13j shows the output noises or errors of the motorcycle's longitudinal speed. It is very compacted between −0.07 and −0.15 s due to the small difference in longitudinal speeds between the actual and estimated states. Figure 13k,l shows the velocity errors of the rear and front wheels, respectively. These errors are very sharp, unlike the longitudinal speed error. They are the result of processing noises and measured noises processed by the extended Kalman filter, representing the difference between speeds in actual and estimated states.

### 3.6. Stopping Distance and Stopping Time

A standby T-SGDMRB alone cannot determine the motorcycle's stopping distance and stopping time. Stopping time and stopping distance of wheels are good indicators to

differentiate one braking system from another. They are based on factors including reaction time ($r_t$), vehicle speed before brake application ($V_i$), road friction coefficient ($C$), braking torque, and environmental conditions like air drag. The mentioned factors are the same at the rear wheel and front wheel except for braking torque ($T_b$). The stopping distance and stopping time of the rear wheel and front wheel are different, and they are expressed as follows.

$$S_{dR} = r_d + B_{dR} = r_d + \left(\tfrac{1}{2} m_R V_i^2 / \frac{T_{b_{(R)}}}{w_R}\right) ; S_{tR} = r_t + \sqrt{\frac{B_{dR}}{a_d}}$$
$$S_{dF} = r_d + B_{dF} = r_d + \left(\tfrac{1}{2} m_R V_i^2 / \frac{T_{b_{(F)}}}{w_R}\right); \ \ S_{tF} = r_t + \sqrt{\frac{B_{dF}}{a_d}}$$

(27)

where $S_{dR}$ is the stopping distance of the ear wheel, $S_{tR}$ is the stopping time of the rear wheel. $S_{dF}$ *and* $S_{tF}$ are the stopping distance and time of the front wheel. $B_{dR}$ *and* $B_{dF}$ are the braking distances of the rear and the front wheel. $r_d$ *and* $r_t$ stand for the reaction distance and reaction time. The magnetorheological brakes have a time response of 0.003 *s* [50]. From this response, the selected motorcyclist's perception should be as minimum as possible. The perception or reaction time ($r_t$) taken by a motorcyclist to apply a brake when there is an obstacle varies in ranges of 0.7 *to* 1.5 *s* [51]. The reaction distance ($r_d$) is the product of reaction time and motorcycle speed before brake application. The reaction time of 0.7 *s* is selected to be used in calculations of stopping distance and stopping time.

Figure 14 shows the analytical results of the stopping time and stopping distance for both the front and rear wheels of a motorcycle under braking with the T-GDMRB. The motorcycle was simulated using a dry asphalt road at varying initial speeds and the controlled braking torque was from an extended Kalman filter. A maximum braking torque of 589.2 Nm was applied to the front wheel, while a braking torque of 383.2 Nm was applied to the rear wheel. Braking torque was constant and set to zero during the entire reaction time of 0 to 0.7 s for both wheels at all speeds. With these defined braking torques, a motorcycle riding at 180 km/h can be slowed down with the rear wheel to stop at 249.18 m after a duration of 6 s, while the front wheel takes 5.2 s to stop at 199.65 m. The motorcycle has a longitudinal speed of 120 km/h; its rear wheel can stop after 4.22 s by traveling 118.5 m and its front wheel travels 96.5 m to stop in 3.8 s. Braking the motorcycle at a longitudinal speed of 60 km/h, the front wheel travels 29.47 m to stop in 2.26 s while the rear wheel stops at 2.48 s after traveling 35.46 m. For a motorcycle riding at 20 km/h equipped with this designed MR brake, the rear wheel can stop after 1.29 s, moving 6.44 m after applying the brake, while the front wheel moves 5.92 m to stop at 1.22 s. This suggests that the higher the riding speed, the longer the duration required to stop the motorcycle, and the longer the stopping distance after activating the brakes.

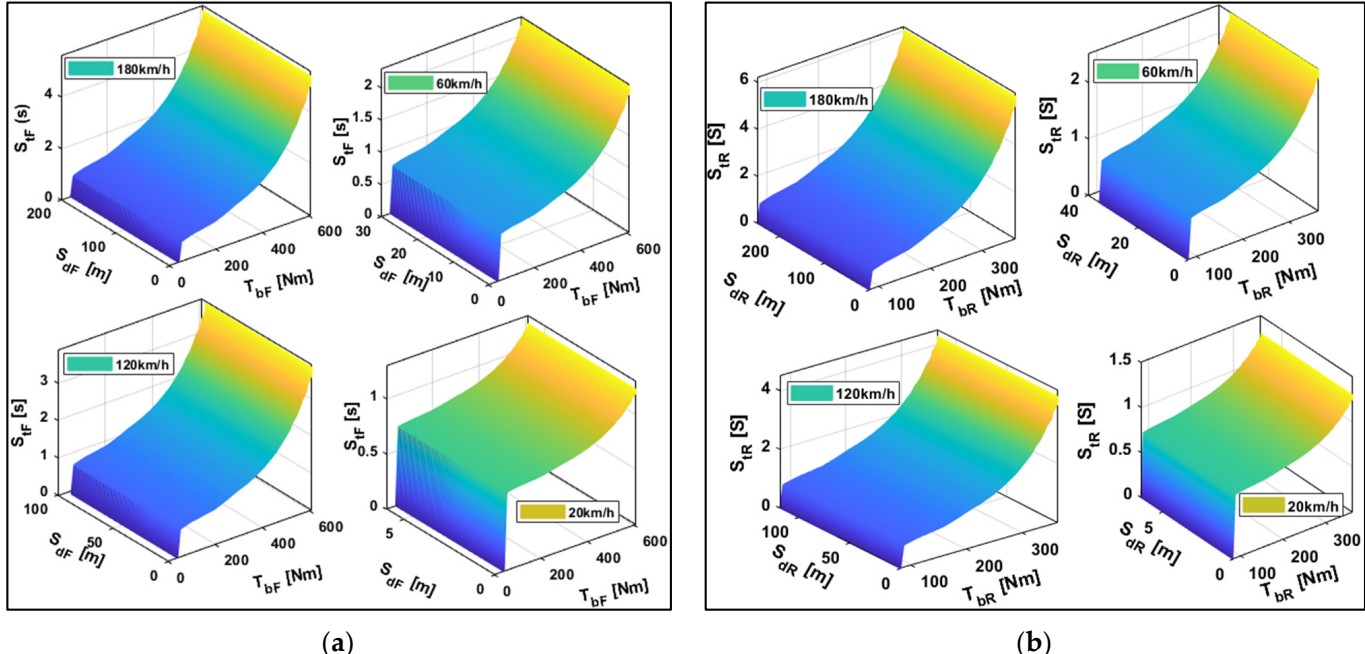

(**a**)                                                                       (**b**)

**Figure 14.** Stopping time and stopping distance of the motorcycle under braking with MR brake. (**a**) Front wheel; (**b**) Rear wheel.

## 4. Conclusions

In this study, the optimal T-shape grooved disc magnetorheological fluid brake is designed to perform the braking of a motorcycle. The braking torque calculations use the Bingham plastic model for MR fluids. The model considers the radial and annular MR fluid gaps, which are between the disc and stator and between the grooved part of the disc and the bobbin. The finite element analysis has been performed with COMSOL Multiphysics to evaluate the magnetic field density generated in the MR fluid concerning the supplied electric current. The design optimization for the geometric parameters of the MR brake has been done by using multi-objective optimization with a genetic algorithm (GA) method. The purpose of conducting optimization was to increase the braking torque applied on the motorcycle (BMW R1200RT) while minimizing the weight of the MR brake and the generated heat during braking. The optimized MR brake weighs 6.1 kg and can generate a maximum torque of 303.9 Nm using an electric current of 2 A at the highest operating temperature of 89.5 °C. Therefore, the performance of the optimally designed MR brake has been evaluated using a simulation approach with MATLAB software. During the simulation, road conditions and the dynamic behaviors of the motorcycle were considered, and an extended-Kalman-filter-based wheel slip control was implemented. Using two MR brakes with a combined braking torque of 589.2 Nm on the front wheel and a single MR brake with 383.2 Nm on the rear wheel, the rear wheel experiences a higher slip rate compared to the front wheel. The front wheel stops earlier than the rear wheel at any riding speed. Simulation of a motorcycle riding at 180 km/h, the front wheel stops after traveling 199.65 m in 5.2 s, while the rear wheel stops after traveling 249.18 m in 6 s. Future work will cover the experiments and practical implementation of this MR brake to the real motorcycle to validate the simulation results and identify the potential improvements to the braking system with magnetorheological fluids.

**Author Contributions:** Conceptualization, P.T. and J.W.S.; software, P.T.; formal analysis, P.T.; investigation, P.T.; writing—original draft preparation, P.T.; writing—review and editing, J.W.S.; supervision, J.W.S. All authors have read and agreed to the published version of the manuscript.

**Funding:** This work was supported by the National Research Foundation of Korea (NRF) grant funded by the Korea government (Ministry of Science and ICT, MSIT) (No. 2023R1A2C1007973). This research was supported by the MSIT (Ministry of Science and ICT), Korea, under the Grand Information Technology Research Center support program (IITP-2023-2020-0-01612) supervised by the IITP (Institute for Information & communications Technology Planning & Evaluation).

**Data Availability Statement:** Not applicable.

**Conflicts of Interest:** The authors declare no conflict of interest.

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
