# Peer review of "Optimal Design and Control Performance Evaluation of a Magnetorheological Fluid Brake Featuring a T-Shape Grooved Disc"

_actuators, doi:10.3390/act12080315_

Round 1

Reviewer 1 Report

This paper aims to determine a magneto-rheological (MR) fluid brake's optimal design and control performance featuring a T-shape grooved disc. The proposed model utilizes mathematical modeling and finite element analysis by commercial software. The optimal design of the MR brake is determined using multi-objective optimization with a genetic algorithm that maximizes braking torque while minimizing the weight of the MR brake and cruising temperature.  The topic is relevant to the field. The manuscript is flawless in its writing, all the relevant literature is cited properly, and the analysis and discussion is thorough. Thus, I recommend that this paper can be accepted in its current form.

Author Response

Please find the attached author's reply.

Reviewer 2 Report

The manuscript reports on a magnetorheological fluid brake's optimal design and control performance featuring a T-shape grooved disc. The simulation results show that the T-shape grooved disc MR brake design enhances the required braking torque to slow down or stop a motorcycle. However, I would recommend the authors to include further information and revise the manuscript based on the following suggestions. My main comments are:

- the abstract lacks in clarity and it should be reshaped to clearly state the framework, the main achievements and to briefly discuss the relevance of the topic in the field.

- there is no discussion about real experimental validation. I would suggest authors to provide more information on such an important theme in experimental validation. This is an important point, only simulation experiment is not enough.

- More importantly, the innovation of this paper is not clearly stated in the summary of the paper, nor is it clearly stated in the abstract.

I would recommend having a native English speaker proofreading the manuscript.

Author Response

(The authors gave the same response as above.)

Reviewer 3 Report

*) In the text there are some typos that should be removed.

*) The abstract appears extensive and reports redundant information but limited to qualitative results. Please, reduce its extent by reporting also a brief summary of the most important quantitative results.

*) Concerning Figure 5, the caption should be self-explanatory.

*) Figures 6(c) and 6(d) are unclear. If possible, improve their quality.

*) The temperature distribution in the MR fluid brake studied in this paper is certainly interesting. However, it is worth noting that the brake (as well as any MR fluid dampers) could be used at different latitudes. In other words, the studied model should take into account such environmental conditions through initial and boundary conditions that simulate, for example, high temperatures of the equatorial zones as well as cold emperatures of the polar zones. So, I think it is useful to specify this need in the text (without modifying the model) by at least a sentence highlighting this possibility listing in the bibliography the following relevant work:

doi: 10.1109/TMAG.2020.3032892

The paper that I recommend to include in the bibliography, even if it studies an MR fluid Damper for the automotive industry, details the study of performance also as a function o the external temperatures to which the vehicle could be subjected.

*) How would the model be affected depending on road wear conditions? Please, if possible, fill this gap.

Author Response

(The authors gave the same response as above.)

Round 2

Reviewer 2 Report

The author has addressed the issues in the paper and provided reasonable explanations for the experimental part and novelty. 

Reviewer 3 Report

All suggestions have been implemented. Therefore, the paper deserves publication.